# Levels of Polychlorinated Dibenzo-*p*-Dioxins/Furans (PCDD/Fs) and Dioxin-Like Polychlorinated Biphenyls (DL-PCBs) in Human Breast Milk in Chile: A Pilot Study

**DOI:** 10.3390/ijerph18094825

**Published:** 2021-04-30

**Authors:** Claudia Foerster, Liliana Zúñiga-Venegas, Pedro Enríquez, Jacqueline Rojas, Claudia Zamora, Ximena Muñoz, Floria Pancetti, María Teresa Muñoz-Quezada, Boris Lucero, Chiara Saracini, Claudio Salas, Sandra Cortés

**Affiliations:** 1Instituto de Ciencias Agroalimentarias, Animales y Ambientales (ICA3), Universidad de O’Higgins, Campus Colchagua, Ruta 90, KM 3, San Fernando 3070000, Chile; claudia.foerster@uoh.cl; 2Laboratorio de Investigaciones Biomédicas, Departamento de Preclínicas, Facultad de Medicina, Universidad Católica del Maule, Talca 3460000, Chile; lzuniga@ucm.cl; 3Centro de Investigaciones y Estudios Avanzados del Maule (CIEAM), Universidad Católica del Maule, Talca 3460000, Chile; csaracini@ucm.cl; 4Laboratorio Química e Inocuidad Alimentaria, Servicio Agrícola Ganadero, Ruta 68 N° 19100, Pudahuel 9020000, Chile; pedro.enriquez@sag.gob.cl (P.E.); jacqueline.rojas@sag.gob.cl (J.R.); claudia.zamora@sag.gob.cl (C.Z.); 5Secretaria Regional de Salud Arica-Parinacota, Maipú 410, Arica 1000000, Chile; ximenadelpilar1967@gmail.com; 6Laboratorio de Neurotoxicología Ambiental, Departamento de Ciencias Biomédicas, Facultad de Medicina, Universidad Católica del Norte, Larrondo N° 1281, Coquimbo 1780000, Chile; pancetti@ucn.cl; 7Centro de Investigación y Desarrollo Tecnológico en Algas y Otros Recursos Biológicos, Universidad Católica del Norte, Larrondo 1281, Coquimbo 1780000, Chile; 8Centro de Investigación en Neuropsicología y Neurociencias Cognitivas, Facultad de Ciencias de la Salud, Universidad Católica del Maule, Talca 3460000, Chile; mtmunoz@ucm.cl (M.T.M.-Q.); blucero@ucm.cl (B.L.); 9Instituto de Investigaciones Agropecuarias INIA Intihuasi, Colina San Joaquín S/N, La Serena 1700000, Chile; claudio.salas@inia.cl; 10Departamento de Salud Pública, Escuela de Medicina, Universidad Pontificia Universidad Católica de Chile, Santiago 8320000, Chile; 11Advanced Center for Chronic Diseases (ACCDIS), Santiago 8320000, Chile; 12Centro de Desarrollo Urbano Sustentable (CEDEUS), Santiago 8320000, Chile

**Keywords:** persistent organic pollutant, dioxins, furans, PCBs, human breast milk, infant exposure, Chile

## Abstract

Persistent organic pollutants (POPs) are organic compounds that resist biochemical degradation, moving long distances across the atmosphere before deposition occurs. Our goal was to provide up-to-date data on the levels of polychlorinated dibenzo-*p*-dioxins/furans (PCDD/Fs) and dioxin-like polychlorinated biphenyls (DL-PCBs) in breast milk from Chilean women and to estimate the exposure of infants due to breast milk consumption. In Chile, we conducted a cross-sectional study based on methodologies proposed by the WHO, with a sample of 30 women recruited from three defined areas: 10 from the Arica Region (urban; Arica and Parinacota Region), 10 from Coltauco (rural; O’Higgins Region), and 10 from Molina (40% rural; Maule Region). High-resolution gas chromatography coupled with high-resolution mass spectrometry (HRGC/HRMS) was performed on pooled samples from each area. We calculated equivalent toxic concentrations (WHO-TEQ) based on the current WHO Toxic Equivalency Factors (TEF). The minimum and maximum values of ∑ PCDDs/Fs + DL-PCBs-TEQ were 4.317 pg TEQ/g fat in Coltauco and 6.31 pg TEQ/g fat in Arica. Molina had a total TEQ of 5.50 pg TEQ/g fat. The contribution of PCDD/Fs was approximately five-fold higher than that of DL-PCBs. The Estimated Daily Intake (EDI) of ∑ PCDDs/Fs + DL-PCBs based on the three pooled samples ranged between 6.71 and 26.28 pg TEQ/kg body weight (bw)/day, with a mean intake of 16.11 (±6.71) pg TEQ/kg bw/day in breastfed children from 0 to 24 months old. These levels were lower than those reported in international studies. Despite the fact that the observed levels were low compared to those in most industrialized countries, the detection of a variety of POPs in breast milk from Chilean women indicates the need for follow-up studies to determine whether such exposures during childhood could represent a health risk in adulthood.

## 1. Introduction

Persistent organic pollutants (POPs) are organic compounds that, to a varying degree, resist photolytic, biological, and chemical degradation. POPs are often halogenated and characterized by low water solubility and high fat solubility, leading to their bioaccumulation in fatty tissues. They are also semi-volatile, enabling them to move long distances in the atmosphere before deposition occurs [1]; indeed, POPs have been detected throughout the world, even in places where they have never been used, such as the Arctic regions. Thus, they are considered the most persistent bioaccumulative and toxic substances in the world [2]. Due to their prolonged half-life and fat solubility, POPs tend to bioaccumulate in animals, especially in the largest species at the top of the food chain. As they are inevitably present in the human body and are found in residual concentrations in breast milk, there are concerns about infants’ exposure. The chronic exposure of humans to POPs is a major health concern as it can have a wide variety of harmful consequences, including reproductive and developmental effects [3,4], neurological and behavioral effects [5,6] or autism [7,8], dermal toxicity [9], and carcinogenicity [10], in addition to metabolic syndrome and obesity [11]. 

POPs are produced and introduced into the environment voluntarily or involuntarily. Variations in environmental conditions, such as those produced by climate change (e.g., an increase in temperature, UV-B radiation), are likely to influence the fate and behavior of POPs, ultimately affecting human exposure [12]. Among the most frequently detected and dangerous POPs for human health are organochlorinated pesticides, such as DDT; industrial chemicals, particularly the dioxin-like polychlorinated biphenyls (DL-PCBs); and industrial by-products, especially polychlorinated dibenzodioxins (PCDD) and polychlorinated dibenzofurans (PCDF) [13]. 

For more than 25 years, the Global Environment Monitoring System of the World Health Organization (GEMS/WHO) Program has collected data on some POPs in foods, including breast milk. In 1998, GEMS/Food published a health assessment of some organochlorine contaminants found in breast milk. In addition, WHO has coordinated three special studies on PCDD/F and DL-PCBs in breast milk, covering 1987–1988, 1992–1993, and 2000–2003. To ensure the reliability and improve the comparability of such data, the WHO conducts regular analytical quality assurance studies for POPs between laboratories [14]. 

This project is part of a WHO/UNEP (United Nations Environment Program) regional program [15] that also covers Argentina, Uruguay, Bolivia, Colombia, Ecuador, Costa Rica, Guatemala, Mexico, Paraguay, and the Dominican Republic. The different studied countries have diverse food consumption patterns and varied exposure to environmental conditions and methods. The project provides national and regional information on the prevalence of POPs in breast milk, considering them as biomarkers of exposure, independent of the environmental burden of such compounds in each country [16]. This publication aims to present up-to-date data on the levels of PCDD/F and DL-PCBs in breast milk in the Chilean population and to estimate the exposure of infants to these substances through breast milk consumption.

## 2. Material and Methods

We conducted a cross-sectional pilot study, using validated protocols suggested by the WHO [14], in a collaborative effort among national universities and the Chilean Agricultural and Livestock Service (*Servicio Agrícola Ganadero*, SAG, Santiago, Chile). 

### 2.1. Study Area and Population

Three sites across Chile were selected in the north (Arica, latitude S18°28′28.56″ and longitude O70°17′52.51″, Arica y Parinacota Region) and two in the south-central area of the country: Coltauco (latitude S34°15′35.96″ and longitude O71°4′40.44″, O’Higgins Region) and Molina (latitude S35°21′12.13″ and longitude O70°54′34.34″, Maule Region) (see graphical abstract). We selected these areas because they engaged in agricultural activities and obtaining human milk from primary health centers was more feasible. 

We recruited volunteer, healthy, lactating mothers who were enrolled in the primary care system. All participants provided signed informed consent and received detailed information about the study’s nature and possible results.

### 2.2. Breast Milk Sampling

The study protocol was based on the fourth WHO-coordinated survey methodology of human milk for POPs [14]. The selection of the participants was carried out through convenience sampling. Participants were healthy women who attended a primary health center for a postpartum check-up and consented to participate. 

A total of 30 women from Arica (*n* = 10), Coltauco (*n* = 10), and Molina (*n* = 10) self-collected 100 mL of breast milk directly into a provided glass container and stored it in their home freezer until the health center personnel collected it. In addition, women completed a personal and dietary survey performed by trained health staff [14]. 

The samples were transported from the primary health center by SAG staff in cold storage and maintained at −20 °C until analysis.

### 2.3. Laboratory Analyses

The samples from each city were pooled and analyzed in the Laboratory of Veterinary Pharmacology of Universidad de Chile (accredited by the National Normalization Institute under the ISO 17025 standard). The informed method of analysis was based on European Regulation N° 709/2014 and N° 2017/644 [17,18], using high-resolution gas chromatography coupled with a high-resolution mass spectrometer (HRGC/HRMS). PCDD, PCDF, and DL-PCBs in breast milk concentrations were reported as ranges and expressed as pg/g of fat or ng/g of fat. The results of equivalent toxic concentrations (WHO-TEQ) were calculated based on the current WHO Toxic Equivalency Factors (TEF) [16]. The informed detection limit (DL) of the method and the percentage of recovery are detailed in Appendix A.

### 2.4. Estimated Daily Intake (EDI) for Infants

Dietary exposure could be indirectly assessed by the calculation of an estimated daily intake (EDI). The EDI combines food consumption data with data on the concentration of chemicals in food, adjusted by the weight of the participants. Estimates can be assessed by a probabilistic approach, where all data available are used, or by a deterministic or point estimate approach, where a mean or median value of the parameters may be used.

An EDI in pg/kg/day was calculated via the deterministic approach for different infant ages and places of sampling, based on Equation (1), with the following assumptions:
The daily infant intake of breast milk was 780 mL at two months old or less; 900 mL at 3–4 months old; 930 mL at 5–6 months old; 600 mL at seven months old or more [19].According to the WHO child growth standards for boys and girls, applied by the Chilean Ministry of Health, the median infant weight (kg) was 4.6 kg for children of 2 months old or less; 6.4 kg for children of 3–4 months old; 7.3 for children of 5–6 months old; 8.7 kg for children of 7–12 months old, and 11.8 for children of 13 months old or more [20].We assumed an absorption efficiency of PCBs and PCDD/Fs in the gastrointestinal tract of 95% [21].

(1)EDI=C × V × F ×0.95
where C is the concentration of PCDD, PCDF, and DL-PCBs in the pooled sample, expressed in pg TEQ/g of fat; V is the volume of daily breast milk intake adjusted to body weight (bw) per group of age expressed in g/kg bw; and F is the breast milk fat content, expressed in g/100 g of milk [8]. An estimation considering the average fat content was also made.

Additionally, a probabilistic approach was adopted for Chilean infants and the sum of PCDD/F + DL-PCBs, assuming a triangular distribution with the minimum, maximum, and most probable values of each parameter.

### 2.5. Statistical Analysis and Modelling

Descriptive analysis of the available data was performed using the IBM SPSS Statistics Base 17.0 software (PASW Statistics, Chicago, IL, USA). Differences between demographics, lifestyles, and mothers’ anthropometric characteristics were assessed by one way ANOVA and χ2 test, with a *p*-value of 0.05. For the concentration of PCDD, PCDF, and DL-PCBs, descriptive analyses were performed, using the congener name of each compound. The values of participants from each city were given by the laboratory as pooled data. The equivalent toxic concentrations (WHO-TEQ) of dioxin-like polychlorinated biphenyls (DL-PCBs), polychlorinated dibenzodioxins (PCDD), and polychlorinated dibenzofurans (PCDF) in the pooled samples were calculated based on WHO Toxic Equivalency Factors (TEF) (2005).

Regarding probabilistic models of PDI, each variable was sampled using the hypercube Latin sampling method, and variables were utilized in a Monte Carlo simulation, using the @Risk 7.5 software (Palisade, Australia) with 100,000 iterations.

## 3. Results

In general terms, the three groups of ten mothers who donated milk samples were young women, with similar age averages at the beginning of pregnancy, ranging from 24.5 ± 5.8 years in Coltauco to 28.9 ± 5.3 years in Arica. The mean BMI in both pre- and post-pregnancy periods was relatively stable, with a more marked weight loss observed in participants from Molina. The BMI values for the whole sample and for both periods were classified as overweight (25.0 to 29.9 kg/m^2^). Regarding the breastfeeding period, the Coltauco sample demonstrated the longest lactation period (11.7 ± 7.9 months) compared to the Arica and Molina samples (4.8 ± 2.7 and 3.4 ± 2.2 months, respectively; *p*-value < 0.01) (Table 1). 

Concerning the total (∑ PCDDs/Fs + DL-PCBs) WHO-TEQ concentrations of the pooled samples, differences were observed between the three locations, with participants from Coltauco displaying the lowest concentrations, with 4.317 pg TEQ/g of fat, and those from Arica displaying the highest, with 6.315 pg TEQ/g of fat (Table 2). Despite these differences, because of the variation in the fat content in the breast milk samples, the total EDI in infants was higher in Coltauco (Table 2). When the EDI was estimated with a mean fat content of 2.75 g/100 mL, the EDI ranged from 8.83 to 29.45 pg TEQ/kg bw/day in Arica, 6.04 to 20.13 pg TEQ/kg bw/day in Coltauco, and 8.05 to 22.66 pg TEQ/kg bw/day in Molina (Table 2). The highest exposure was in infants aged 0 to 2 months old from Arica, with 29.45 pg total TEQ/kg bw/day (Appendix A). The Chilean infant’s daily intake of ∑ PCDD/Fs + DL-PCBs, as estimated using the probabilistic approach, was, on average, 14.21 (±3.61) pg TEQ/kg bw/day, with a P95% of 20.90 pg TEQ/kg bw/day (Appendix A).

## 4. Discussion

In the present study, the minimum and maximum detected values of total-TEQ (∑ PCDDs/Fs + DL-PCBs) were between 4.32 pg TEQ/g of fat and 6.32 pg TEQ/g of fat (mean = 5.38 pg TEQ/g of fat), and the contribution of PCDD/Fs was approximately five-fold higher than that of PCBs. It is important to note that these levels are lower than those given by van den Berg et al. [22], who reported (in WHO-TEQ 2005) [16] values of approximately 2 pg/g of fat for DL-PCBs and 8.5 pg/g of fat for PCDD/Fs in the 4/5th surveys of 2005–2008 in Chile [15], with similar contribution ratios. Few studies have provided WHO-TEQ values of PCDDs/Fs and DL-PCBs in pooled samples from the general population. Most values presented in the literature are based on WHO TEF values from 1998. In this context, Table 3 summarizes several studies carried out mainly in China and a couple of countries in Europe, such as Belgium and Sweden, which were based on WHO TEF (2005) in pooled samples, as in our study. Data previously reported for Chile revealed similar PCDDs/Fs + DL-PCBs levels to those reported for China’s six counties, which were, on average, 9.09 pg TEQ/g fat [23]. Furthermore, according to Fang et al. [24], both Brazil and Chile displayed higher total WHO-TEQ values than African countries, although current data from samples taken in 2019 show similar levels to those reported for Sweden in 2013, reaching a total level of PCDDs/Fs + DL-PCBs of between 4.6 and 5.5 pg TEQ/g of fat [24].

The EDIs of ∑PCDD/F + DL-PCBs based on the three pooled samples ranged between 6.71 and 26.28 pg TEQ/kg bw/day, with a mean intake of 16.11 ± 6.71 pg TEQ/kg bw/day in children aged 0 to 24 months old. This estimated mean is lower than that found in studies conducted in developed countries [28,29,30,31]. The EDI was slightly different between towns due to differences in the fat content. The highest exposures found in infants of 0–2 months old from Coltauco (26.28 pg total TEQ/kg bw/day) and Arica, with a mean fat content of 2.58% (29.45 pg total TEQ/kg bw/day) (Appendix A), are far below those reported by Lu et al. [32], who estimated mean EDIs of 54–57 pg TEQ/kg bw/day in infants from China, but higher than those found in Hungary during the first 84 days of life, where the maximum EDI was estimated at 16.54 ± 13.02 pg TEQ/kg bw/day [21]. An explanation for these exposures is that Arica, Coltauco, and Molina experience significant exposure to pesticides (not POPs), but they are not industrialized cities.

When compared to the tolerable daily intake (TDI), EDIs were higher than the suggested values given by the World Health Organization (WHO) of 1–4 pg TEQ/kg bw per day [33]. These safety standards are intended for chronic lifetime exposure; thus, they are not directly applicable to breastfeeding, which covers a much shorter period of life [27]. Despite this, and although there is not enough evidence of an association between this TDI exceedance during lactation and health outcomes in the future [15], some studies suggest that perinatal background exposure to PCBs and PCDD/Fs might be associated with a greater susceptibility to infectious diseases [34]. It is also related to disturbances in the thyroid hormone endocrine balance of infants and children [35] and their mothers [36,37], among other effects.

This study provides up-to-date data on the levels of POPs in infants from different geographical areas of Chile, with diverse socio-cultural and environmental characteristics. Although this study presents new information about PCDDs/F and DL-PCB levels in human milk in the Chilean population, the results must be interpreted with caution as they represent only a small part of the population. In this regard, the main limitations of this pilot study were the low number of samples per city and the impossibility of an individual analysis of the breast milk samples. However, our results complement previous preliminary national data regarding the actual low environmental exposure to these toxic compounds and reinforce the need to maintain its surveillance at the country level. 

## 5. Conclusions

Ten years after the first report was published on PCDDs/F and DL-PCB levels in human milk in the Chilean population, an approximately 50% decrease in these levels has been observed. In Arica, participants, who were from urban environments, presented the highest levels of PCDDs/Fs and DL-PCBs in human milk, whereas the rural participants from Coltauco showed the lowest amounts. Arica is a mining region and a more industrialized city than Coltauco and Molina, which could explain these results. In this regard, further research should measure such levels in the most industrialized cities of Chile, such as Santiago.

Based on this study, a strengthened surveillance system for the presence of POPs in maternal milk should be implemented at the country level to promote strategies that reduce children’s potential exposure and mitigate the health risk at this early age.

Despite the fact that the observed levels were low, the detection of a variety of POPs in breast milk in Chilean women highlights the need to maintain efforts to reduce exposure in infants via breast milk consumption and develop more in-depth studies to evaluate chronic effects on health in adulthood.

## Figures and Tables

**Table 1 ijerph-18-04825-t001:** Demographic information for pooled samples of women from 3 regions of Chile.

Mother Characteristics	Arica	Coltauco	Molina
Donors (*n*)	10	10	10
Age at start of pregnancy (mean ± ED)	28.9 ± 5.3	24.5 ± 5.8	26.3 ± 5.7
Pre-pregnancy body mass index (kg/m^2^)	26.7 ± 4.0	26.5 ± 3.9	30.3 ± 4.0
Post-pregnancy body mass index (kg/m^2^)	25.9 ± 4.5	27.4 ±5.0	28.4 ± 3.4
Breastfeeding period **	4.8 ±2.7	11.7 ± 7.9	3.4 ± 2.2
Current residence (rural)	0 (0%)	9 (90%)	4 (40%)
Last 10 years residence (rural)	0 (0%)	10 (100%)	4 (40%)

** one-way ANOVA (*p* = 0.002).

**Table 2 ijerph-18-04825-t002:** The equivalent toxic concentrations (WHO-TEQ) of dioxin-like polychlorinated biphenyls (DL-PCBs), polychlorinated dibenzodioxins (PCDD), and polychlorinated dibenzofurans (PCDF) of the pooled samples and estimated daily intakes (EDI) of infants in 3 regions of Chile.

City	Content of Fat g/100 mL	TEQ-PCDDs/Fs pg/g Fat	TEQ-DL-PCBs pg/g Fat	TEQ-Total pg/g Fat	EDI (Total) of Infants in pgTEQ/kg Body Weight (bw)	EDI (Total) of Infants in pgTEQ/kg bw (Mean Fat: 2.75 g/100 mL)
Arica	2.09	5.167	1.148	6.315	6.71–22.38	8.83–29.45
Coltauco	3.59	3.621	0.696	4.317	7.88–26.28	6.04–20.13
Molina	2.58	4.612	0.891	5.503	7.56–24.07	8.05–22.66

The values were calculated based on WHO Toxic Equivalency Factors (TEF) (2005).

**Table 3 ijerph-18-04825-t003:** Levels of dioxin-like polychlorinated biphenyls (DL-PCBs), polychlorinated dibenzodioxins (PCDD), and polychlorinated dibenzofurans (PCDF) in human milk in similar studies.

Country	Pools/Simples	Year of Sampling	∑ PCDDs/Fs	∑ DL-PCBs	∑ PCDDs/Fs + DL-PCBs	Reference
Chile	3 pools/30 samples	2019	3.6–5.2	0.7–1.2	4.3–6.3	Present study
Chile	1 pool/NA	2007–2009	NA	NA	9.7	[15]
China	6 pools/179 samples	2017–2018	6.96	2.13	9.09	[23]
China	32 pools/1760 samples	2011	NA	NA	2.4–12.8	[25]
Sweden	8 pools / 79 samples	2008–2011	2.7–3.1	1.9–2.4	4.6–5.5	[24]
Belgium	1 pool / 84 samples	2009–2010	6.9	3.7	10.7	[26]
China	24 pools/1237 samples	2007	1.38–5.82	0.56–2.93	2.12–8.61	[27]

The values were calculated based on WHO Toxic Equivalency Factors (TEF) (2005).

## Data Availability

The data presented in this study are available in the Appendix A.

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
