# Peer review of "Levels of Polychlorinated Dibenzo-p-Dioxins/Furans (PCDD/Fs) and Dioxin-Like Polychlorinated Biphenyls (DL-PCBs) in Human Breast Milk in Chile: A Pilot Study"

_ijerph, 2021, doi:10.3390/ijerph18094825_

Round 1

Reviewer 1 Report

This paper provides an updated data on the exposure of PCDD/Fs and DL-PCBs in Chile. The results are novel despite the very limited sample size applied.

  1. Since a quite small sample size was used in this study, how did the authors make sure that the samples are representative of the population?
  2. I wonder why did the authors use more than three significant digits in the manuscript. In my opinion, three significant digits is enough.
  3. Results and Discussion: Some publications provided the latest data on the WHO-TEQ levels of PCDD/Fs and DL-PCBs in some other regions, although not in pooled samples (Germany: Li ZM, et al. (2019) Environmental Science & Technology 54(2): 1111-1119. Uganda: Matovu, H., et al. (2021). Science of the Total Environment 770: 145262.) I would suggest to compare the results.
  4. There are some typing errors in the text. The authors should check carefully.

Author Response

Reviewer 1:

His paper provides an updated data on the exposure of PCDD/Fs and DL-PCBs in Chile. The results are novel despite the very limited sample size applied.

Since a quite small sample size was used in this study, how did the authors make sure that the samples are representative of the population?

R: Because of the limited sampling, this study was not intended to be representative of the population, but an exploratory pilot study for identifying future research directions in research. This was accounted for in the limitations of the study.

I wonder why did the authors use more than three significant digits in the manuscript. In my opinion, three significant digits is enough.

R: Thank you for the observation, a maximum of three significant digits is incorporated in the manuscript. The exception was the calculated pg TEQ WHO/g, based on the levels found in pg/g analyzed and the WHO Toxic Equivalency Factors (TEF) (2005), included in the supplementary material.

Results and Discussion: Some publications provided the latest data on the WHO-TEQ levels of PCDD/Fs and DL-PCBs in some other regions, although not in pooled samples (Germany: Li ZM, et al. (2019) Environmental Science & Technology 54(2): 1111-1119. Uganda: Matovu, H., et al. (2021). Science of the Total Environment 770: 145262.) I would suggest to compare the results.

R: Thank you for the suggestions, were very interesting and were incorporated in the health effects paragraph in the discussion. 

There are some typing errors in the text. The authors should check carefully.

R: Some mistakes were solved. Also, the manuscript was reviewed by the English Editing Services of MDPI.

Reviewer 2 Report

This study was conducted well. The data are readily understandable and straightforward. The manuscript is written in a clear and concise manner. I have no criticisms and recommend publication.

This paper is describes analytical measurement of the dioxins and doxin-like compounds in breast milk from Chilean mothers. The TEQ of dioxins and dioxin-like chemicals is declining in this population, similar to what is being reported in other countries. There is no hypothesis testing, it’s a simple “measure and report the results” paper. The methods and statistical analysis used are standard, as this is part of a larger WHO study to examine these chemicals in the environment and in people. It isn’t a revolutionary paper, but it constitutes a piece of information that is important for risk assessment and public health. Since it involves breastfeeding mothers, it fits within the “Women’s Health” sub-category. I read the paper and found it to be in good shape. I did not find any mistakes or things to critique, so I have recommended publication as is. 

Author Response

This study was conducted well. The data are readily understandable and straightforward. The manuscript is written clearly and concisely. I have no criticisms and recommend publication.

This paper describes analytical measurement of the dioxins and dioxin-like compounds in breast milk from Chilean mothers. The TEQ of dioxins and dioxin-like chemicals is declining in this population, similar to what is being reported in other countries. There is no hypothesis testing, it’s a simple “measure and report the results” paper. The methods and statistical analysis used are standard, as this is part of a larger WHO study to examine these chemicals in the environment and in people. It isn’t a revolutionary paper, but it constitutes a piece of information that is important for risk assessment and public health. Since it involves breastfeeding mothers, it fits within the “Women’s Health” sub-category. I read the paper and found it to be in good shape. I did not find any mistakes or things to critique, so I have recommended publication as is.

R: Thank you very much for your comments.

Reviewer 3 Report

The article entitled "Levels of polychlorinated dibenzo-p-dioxins/furans (PCDD/Fs) and dioxin-like polychlorinated biphenyls (DL-PCBs) in human breast milk in Chile: a pilot study" was designed to assess POP concentration in human milk from 3 different regions in Chile. The authors used a very small number of samples, thus calling it a pilot study.

I recommend that the authors use some type of English editing service to make the manuscript more readable with smooth transitions. 

The authors used methodologies that appear to be the norm for these types of studies which include the pooling of the samples for the chemical analyses, and deterministic and probabilistic approaches to calculate the individual intake.   For readers not familiar with the approaches, it would be best to briefly explain what they are.

Here are other areas that can be improved:

1) Add information on congeners in the introduction if the authors still want to describe them in the results section. I am not sure if that information complemented the findings presented and conclusions.

2) a statistical section to the Materials and Methods

2) Separate Results from Discussion

3) to the discussion add a paragraph specific on the limitations of the study

Author Response

I recommend that the authors use some type of English editing service to make the manuscript more readable with smooth transitions.

R: The manuscript was reviewed by the English Editing Services of MDPI.

The authors used methodologies that appear to be the norm for these types of studies which include the pooling of the samples for the chemical analyses, and deterministic and probabilistic approaches to calculate the individual intake. For readers not familiar with the approaches, it would be best to briefly explain what they are.

R: Thanks for the suggestion. An explanation of the approaches was added in the Materials and Methods section 2.5. Estimated daily intakes (EDI) for infants (first paragraph).

Here are other areas that can be improved:

1) Add information on congeners in the introduction if the authors still want to describe them in the results section. I am not sure if that information complemented the findings presented and conclusions.

R: Thank you for the annotation, the paragraph was erased because the table is in the supplementary material.

2) a statistical section to the Materials and Methods

R: Done (2.6. Statistical analysis and modeling)

2) Separate Results from Discussion

R: Done (4. Discussion)

3) to the discussion add a paragraph specific on the limitations of the study

R: Added in the last paragraph of the Discussion section.